

# Antibiotic resistance profile of *Helicobacter pylori* to 14 antibiotics: a multicenter study in Fujian, China

Xueping Huang[1,2,*], Baihe Wu[3,*], Qiuzhao Chen[4], Yushan Chen[5], Xinhua Ji[6], Xiang Zhou[6], Biao Suo[4], Zhihui Lin[1,2] and Xiaoling Zheng[7,8]

[1] Department of Gastroenterology, Shengli Clinical Medical College of Fujian Medical University, Fuzhou, Fujian, China
[2] Department of Gastroenterology, Fujian Provincial Hospital, Fuzhou, Fujian, China
[3] Department of Gastroenterology, Fifth Affiliated Hospital of Sun Yat-sen University, Zhuhai, Guangdong, China
[4] Department of Gastroenterology, Xiamen Hospital of Traditional Chinese Medicine, Xiamen, Fujian, China
[5] Department of Gastroenterology, Fuzhou No.1 Hospital Affiliated with Fujian Medical University, Fuzhou, Fujian, China
[6] Department of Gastroenterology, Pucheng Hospital, Nanping, Fujian, China
[7] Department of Digestive Endoscopy, Shengli Clinical Medical College of Fujian Medical University, Fuzhou, Fujian, China
[8] Department of Digestive Endoscopy, Fujian Provincial Hospital, Fuzhou, Fujian, China
* These authors contributed equally to this work.

Corresponding authors
Zhihui Lin, 13799371368@139.com
Xiaoling Zheng, fjzhengxl@163.com

## ABSTRACT

**Background and aim:** Efficacy of *Helicobacter pylori* (*H. pylori*) eradication is related to the local antimicrobial resistance epidemiology. We aimed to investigate the antibiotic resistance of *H. pylori* in Fujian, China.

**Methods:** *H. pylori*-infected patients in four centers were enrolled in the study from Oct 2019 to Jan 2022. The bacteria were isolated, cultured and identified from the biopsy of patients' gastric mucosa samples. Antimicrobial susceptibility testing was performed by a modified broth microdilution method for *H. pylori* to seven guideline-recommended antibiotics and seven potential choices for *H. pylori* eradication.

**Results:** A total of 205 *H. pylori* strains were isolated. The resistance rates of amoxicillin (AMX), amoxicillin and clavulanate potassium (AMC), cefixime (CFM), gentamicin (GEN), tetracycline (TET), doxycycline (DOX), azithromycin (AZM), clarithromycin (CLR), levofloxacin (LVFX), sparfloxacin (SPFX), metronidazole (MTZ), tinidazole (TID), rifampicin (RFP) and furazolidone (FZD) were 11.22%, 12.20%, 7.32%, 12.20%, 4.88%, 4.39%, 44.39%, 43.90%, 30.24%, 21.46%, 40.98%, 45.85%, 5.37% and 10.24%, respectively. The rates of pan-sensitivity, single, double, triple and multiple resistance for seven guideline-recommended antibiotics were 32.68%, 30.24%, 13.17%, 7.76%, and 14.15%, respectively. The main double-resistance patterns were CLR+MTZ (10/205, 5%) and CLR+LVFX (9/205, 4%). The main triple-resistance pattern was CLR+MTZ+ LVFX (15/205, 7%).

**Conclusions:** In Fujian, the prevalence of *H. pylori* resistance to AZM, CLR, LVFX, SPFX, MTZ, and TID was high, whereas that to AMX, AMC, GEN, CFM, TET, DOX, RFP and FZD was relatively low. CFM and DOX are promising new choices for *H. pylori* eradication.

# INTRODUCTION

*Helicobacter pylori* (*H. pylori*) is a gram-negative bacillus that is estimated to infect almost half of the world's population (*Hooi et al., 2017*). *H. pylori* was categorized as a type 1 carcinogen in 1994 by the *International Agency for Research on Cancer (1994)*. Subsequently, persistent infection has been demonstrated to be associated with the development of severe gastric complications (*Yamaoka, 2018*), such as gastric cancer, peptic ulcer, chronic gastritis, and MALT-lymphoma. The prevalence of *H. pylori* infection in China was high, with a mean prevalence of 55% during 1983–2013 (*Nagy, Johansson & Molloy-Bland, 2016*). Gastric cancer is the second and third most common cancer in men and women, respectively, in China. In 2015, there were 679,100 estimated new cases of gastric cancer in China, which remained the second leading cause of cancer-related deaths in both men and women (*Chen et al., 2015*). Moreover, the occurrence of gastric cancer is concentrated in the central and eastern areas of China (*Sun, Zheng & Zhang, 2019*). Fujian Province is located in the east of China and is one of the regions with the highest incidence in China. Numerous studies (*Lee et al., 2016*; *de Martel et al., 2012*; *Gonzalez et al., 2012*) have shown that up to 90% of gastric cancers can be attributed to *H. pylori* infection; therefore, elimination of this bacterium should be an effective strategy to reduce gastric cancer.

However, the optimization of *H. pylori* eradication therapy remains an ongoing challenge worldwide. The efficacy of clarithromycin-containing regimens has fallen below 80% in many Asian countries (*Nishizawa et al., 2015*; *Chung et al., 2016*), which is unacceptable in clinical practice owing to the rising prevalence of clarithromycin resistance (*Kuo et al., 2017*; *Arslan, Yilmaz & Demiray-Gurbuz, 2017*). Among the reasons for treatment failure, antibiotic resistance is a key element in the treatment of *H. pylori* infection (*Megraud, 2018*). Moreover, *H. pylori* resistance to antibiotics has reached alarming levels worldwide (*Kuo et al., 2017*). Therefore, obtaining information on the local primary antibiotic resistance of *H. pylori* is of great importance in guiding its treatment. In addition, it is critical to seek novel and effective therapeutic drugs for *H. pylori* eradication in an era of increasing antibiotic resistance.

Amoxicillin (AMX), tetracycline (TET), clarithromycin (CLR), levofloxacin (LVFX), metronidazole (MTZ), rifampicin (RFP), and furazolidone (FZD) are seven antibiotics recommended by guidelines (*Malfertheiner et al., 2022*; *Liu et al., 2018*). Up to now, there has been a lot of research on these drugs, but there is still a lack of research on new drugs, such as amoxicillin and clavulanate potassium(AMC), cefixime (CFM), gentamicin (GEN), doxycycline (DOX), azithromycin (AZM), sparfloxacin (SPFX), and tinidazole (TID), which may replace some of those recommended by guidelines. Our study aimed to assess the prevalence of *H. pylori* primary resistance to seven guideline-recommended antibiotics as mentioned above and seven potential choices for the treatment of *H. pylori* in Fujian, China.

## MATERIALS AND METHODS

### Patients

Patients were enrolled in this study from four centers from Oct 2019 to Jan 2022 in Fujian Province, China. They were the Fujian Provincial Hospital (Fuzhou, Fujian, China), Fuzhou First Hospital (Fuzhou, Fujian, China), Xiamen Hospital of Traditional Chinese Medicine (Xiamen, Fujian, China), and Pucheng Hospital (Nanping, Fujian, China). All patients were >18 years of age. Patients who had malignant tumors, endoscopy contraindications, or took antibiotics, proton pump inhibitors (PPI), H2 receptor antagonists, bismuth, and other drugs 4 weeks before enrollment in the study were all excluded from the study. General information of the patients was recorded, including age, sex, endoscopic diagnosis results, and telephone number. This study was approved by the Ethics Committee of the Fujian Provincial Hospital (Approval No. K2019-08-008) and informed consent was obtained from each patient.

### Isolation and identification of *H. pylori* strains

During gastroscopy, a single biopsy specimen was obtained from the antrum of the stomach of each patient to determine the presence of *H. pylori* using a rapid urease test. If the test results were positive, two additional gastric mucosal biopsy specimens were collected from the antrum and *corpus* and stored in a liquid medium containing brain heart immersion broth, 10% calf serum, vancomycin 10 mg/L, methomycin 5 mg/L, cefsulodin 5 mg/L and diclofenac B 5 mg/L (manufactured by Zhuhai Yeoman Bioengineering Products Factory) for *H. pylori* culture. The *H. pylori* isolates were identified by the presence of curved gram-negative rods and positive reactions to the urease, catalase, and oxidase tests.

### Antibiotic susceptibility testing of *H. pylori* strains

Antimicrobial susceptibility testing was performed using the modified broth microdilution method for *H. pylori* strains to 14 antibiotics, including amoxicillin (AMX), amoxicillin and clavulanate potassium (AMC), cefixime (CFM), gentamicin (GEN), tetracycline (TET), doxycycline (DOX), azithromycin (AZM), clarithromycin (CLR), levofloxacin (LVFX), sparfloxacin (SPFX), metronidazole (MTZ), tinidazole (TID), rifampicin (RFP), and furazolidone (FZD). The plates were provided by Zhuhai Yeoman Bioengineering Products Factory. The standard strain ATCC43504 (provided by Shanghai Bioplus Biotechnology Company) was used as a control. Bacterial cultures were diluted in fresh serum-containing brain heart immersion broth at a ratio of 1:100 to produce an inoculum yielding approximately $2 \times 10^6$ colony forming units (CFU)/ml. Two dilution concentrations of each antibiotic were available, one of which was the breakpoint concentration of the antibiotic, and 200 μl of bacterial dilution is added to each well. Separate antibiotic-free medium and antibiotic-free bacteria-containing medium were used as controls. The plates were covered with adhesive seals and incubated at 37 °C for 72 h. A total of 2,3,5-Triphenyltetrazolium chloride (TTC) was used to indicate bacterial growth, where red color indicated bacterial growth and no color change (yellow) indicated no bacterial growth.

Antibiotic powders AMX, CLR, and LVFX were purchased from BioDuly Biotechnology Co. Ltd., (Nanjing, China), and AMC, CFM, GEN, TET, DOX, AZM, SPFX, MTZ, TID, RFP and FZD were from Meilun Biotechnology Co. Ltd. (Dalian, China). If the bacteria grew at the breakpoint concentration of these antibiotics, the bacteria were judged to be resistant to the antibiotic, and conversely are judged to be sensitive. Some breakpoints were established in accordance with the *European Committee on Antimicrobial Susceptibility Testing (EUCAST) (2023)*, which indicated a resistance breakpoint of >8 mg/L for MTZ, >1 mg/L for LEV, and >1 mg/L for RFP. Additionally, previous studies (*Bai et al., 2015*; *Su et al., 2013*; *Ji et al., 2016*) were referenced to determine breakpoints for other antibiotics, including ≥2 mg/L for AMX, >0.5 mg/L for CLA, >1 mg/L for AZM, >1 mg/L for CFM, ≥16 mg/L for GEN, ≥2 mg/L for FZD, and ≥4 mg/L for TET. The study utilized consistent resistance breakpoints for the same class of drugs for the rest of antibiotics, including ≥2 mg/L for AMC, ≥4 mg/L for DOX, >8 mg/L for TID, >1 mg/L for SPFX.

## Statistical analysis

All data in this study were analyzed using R 4.0 (*R Core Team, 2020*). The UpSet plot was used to visualize the intersections of the drug resistance sets. Differences of resistance rates in different age, sex, and disease groups were assessed using the chi-square test, $P < 0.05$ was regarded as statistically significant.

## RESULTS

### Patient characteristics and endoscopic data

A total of 205 *H. pylori* strains were successfully isolated from the biopsy patients' gastric mucosa samples (90 women and 115 men; age range, 18–79 years; mean age, 45 ± 13.47 years). The endoscopic diagnoses were as follows: chronic superficial gastritis, 52/205 (25.4%); chronic atrophic gastritis, 108/205 (52.7%); and peptic ulcer, 45/205 (22%).

### Antibiotic resistance of *H. pylori* to 14 antibiotics

The rates of antibiotic resistance to AMX, AMC, CFM, GEN, TET, DOX, AZM, CLR, LVFX, SPFX, MTZ, TID, RFP, and FZD were 11.22%, 12.20%, 7.32%, 12.20%, 4.88%, 4.39%, 44.39%, 43.90%, 30.24%, 21.46%, 40.98%, 45.85%, 5.37% and 10.24%, respectively (Fig. 1).

### Multiple antibiotic resistance of *H. pylori* to seven antibiotics recommended by guidelines

In our study, only 32.68% of patients were susceptible to all seven guideline-recommended antibiotics. The rates of pan-sensitivity, single resistance, double resistance, triple resistance, and multiple resistance were 32.68%, 30.24%, 13.17%, 7.76%, and 14.15%, respectively (Fig. 2). The resistance patterns are shown in Fig. 3. The main double-resistance patterns were CLR+MTZ (10/205, 5%) and CLR+LVFX (9/205, 4%). The main triple-resistance pattern was CLR+MTZ+LVFX (15/205, 7%).

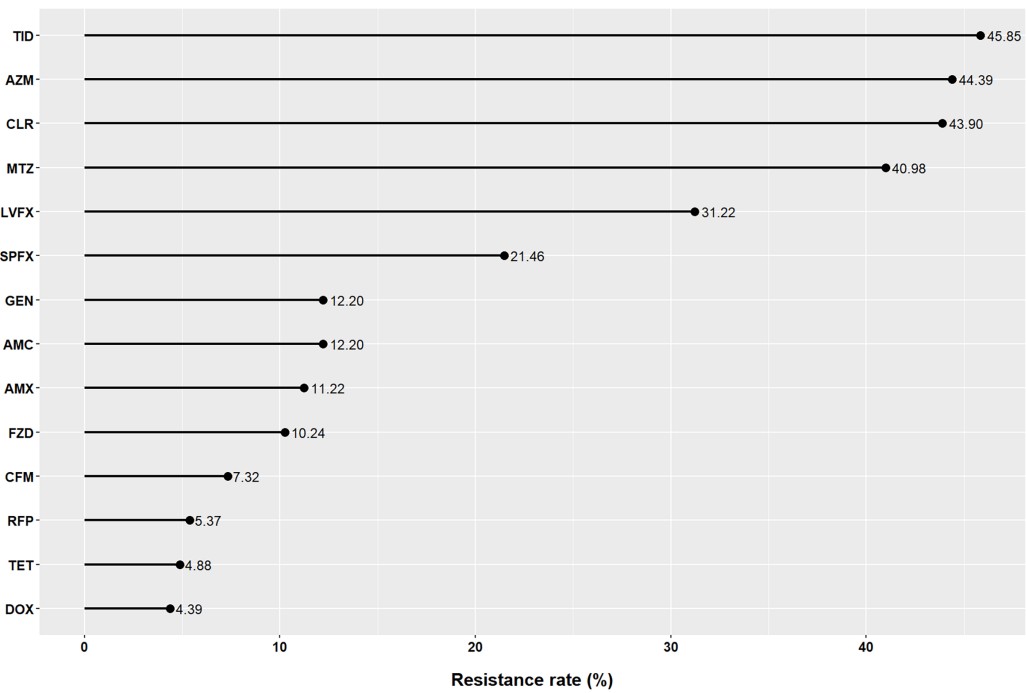

**Figure 1 Resistance of *Helicobacter pylori* to 14 antibiotics in Fujian.** AMX, amoxicillin; AMC, amoxicillin and clavulanate potassium; CFM, cefixime; GEN, gentamicin; TET, tetracycline; DOX, doxycycline; AZM, azithromycin; CLR, clarithromycin; LVFX, levofloxacin; SPFX, sparfloxacin; MTZ, metronidazole; TID, tinidazole; RFP, rifampicin; FZD, furazolidone.

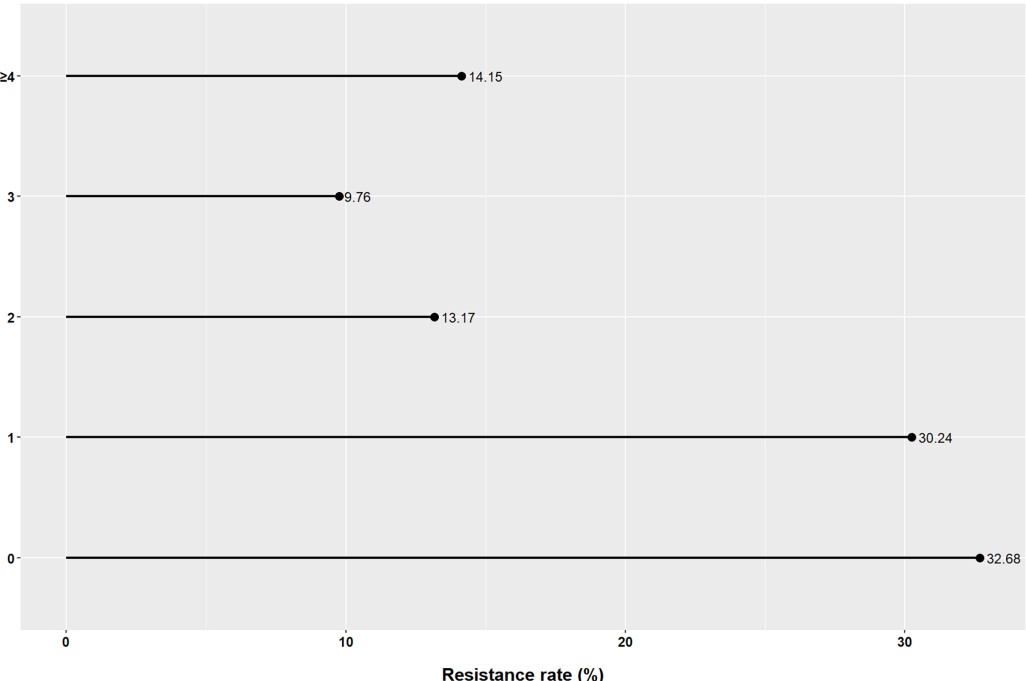

**Figure 2 Multiple resistance of *Helicobacter pylori* to seven guideline-recommended antibiotics in Fujian.** The vertical axis represents the number of co-resistant antibiotics.

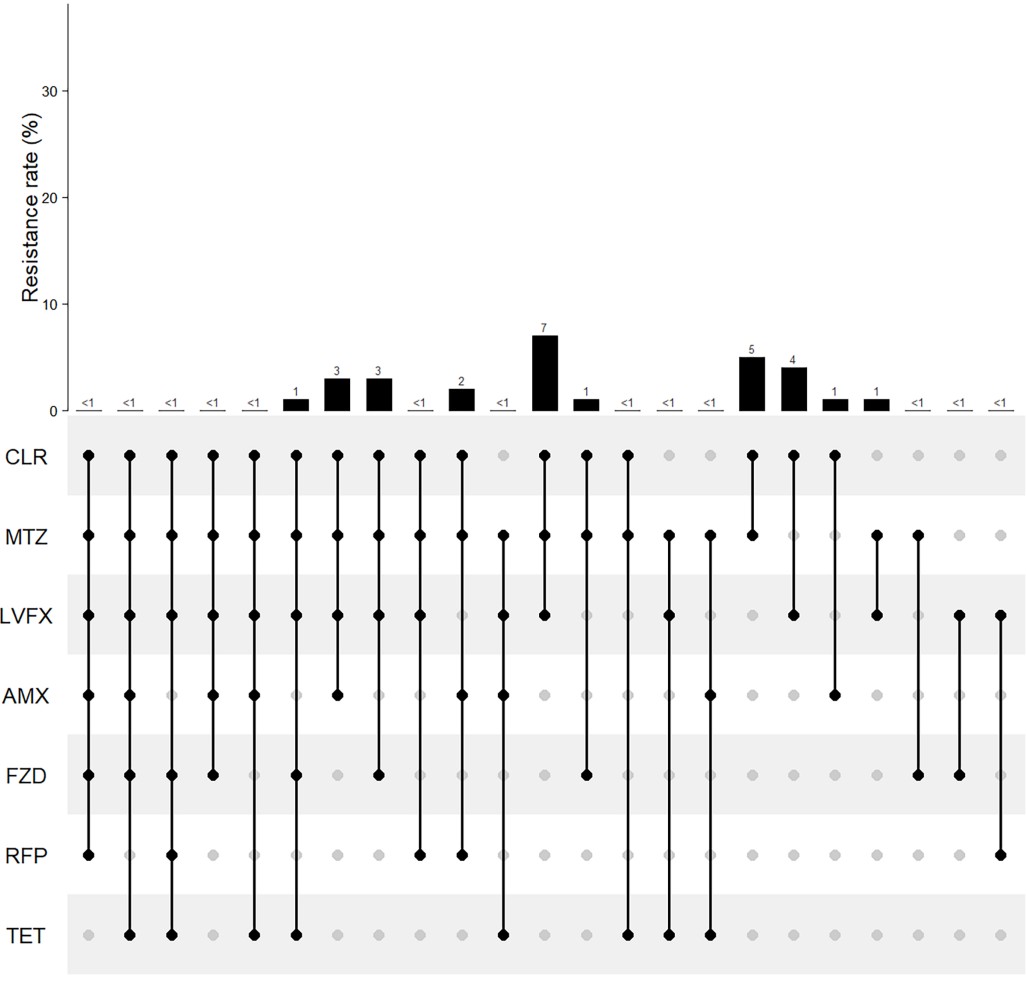

**Figure 3 Multiple resistance patterns of *Helicobacter pylori* to seven guideline-recommended antibiotics in Fujian.** The horizontal axis represents various antibiotic resistance patterns with black dot. AMX, amoxicillin; AMC, amoxicillin and clavulanate potassium; CFM, cefixime; GEN, gentamicin; TET, tetracycline; DOX, doxycycline; AZM, azithromycin; CLR, clarithromycin; LVFX, levofloxacin; SPFX, sparfloxacin; MTZ, metronidazole; TID, tinidazole; RFP, rifampicin; FZD, furazolidone.

## Impact of sex, age and disease patterns on *H. pylori* antibiotic resistance

The AMC resistance rate in the female group was lower than that in the male group (6.67% *vs*. 16.52%, $P = 0.032$). For the other 13 antibiotics, there was no difference in the resistance rates by sex (Fig. 4). No significant differences were detected in antibiotic resistance among the different age groups (Fig. 5). Similarly, there were no statistically significant differences among the different disease patterns (Table 1).

## DISCUSSION

*H. pylori* gastritis should be defined as an infectious disease, irrespective of symptoms and complications, such as peptic ulcers and gastric cancer (*Sugano et al., 2015*).
Antibiotic-based therapies play a key role in the treatment of infectious diseases. However,

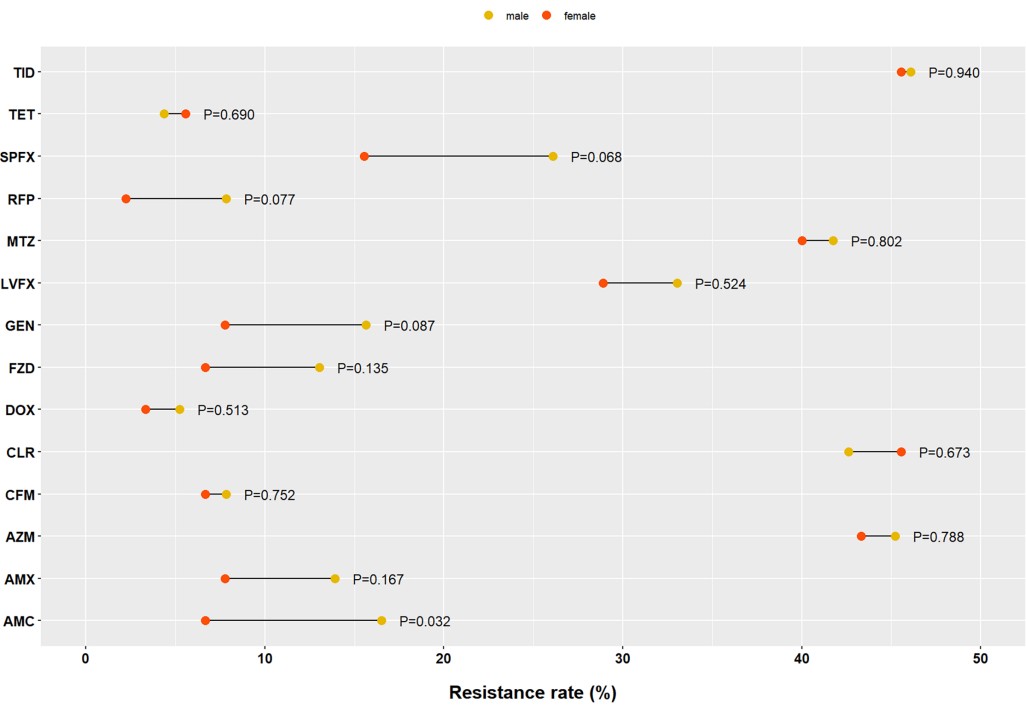

**Figure 4 Impact of sex on *Helicobacter pylori* resistance.** AMX, amoxicillin; AMC, amoxicillin and clavulanate potassium; CFM, cefixime; GEN, gentamicin; TET, tetracycline; DOX, doxycycline; AZM, azithromycin; CLR, clarithromycin; LVFX, levofloxacin; SPFX, sparfloxacin; MTZ, metronidazole; TID, tinidazole; RFP, rifampicin; FZD, furazolidone.

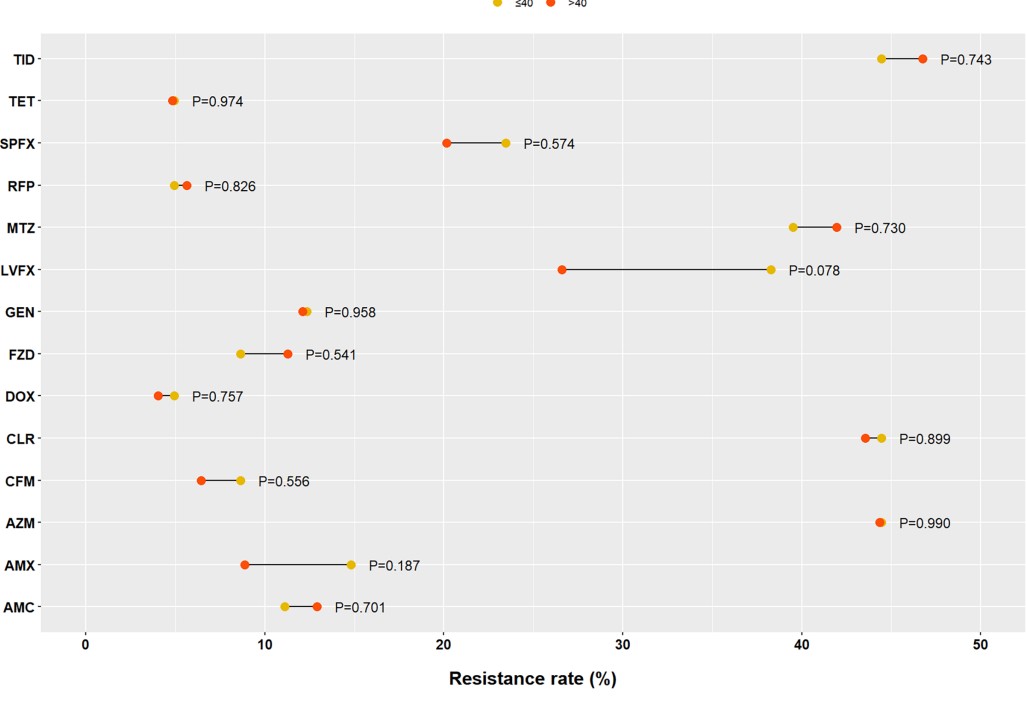

**Figure 5 Impact of age on *Helicobacter pylori* resistance.** AMX, amoxicillin; AMC, amoxicillin and clavulanate potassium; CFM, cefixime; GEN, gentamicin; TET, tetracycline; DOX, doxycycline; AZM, azithromycin; CLR, clarithromycin; LVFX, levofloxacin; SPFX, sparfloxacin; MTZ, metronidazole; TID, tinidazole; RFP, rifampicin; FZD, furazolidone.

**Table 1 Impact of disease patterns on *Helicobacter pylori* resistance.**

| Antibiotics | Diagnosis | | | P |
|---|---|---|---|---|
| | CSG *n*(%) | CAG *n*(%) | PU *n*(%) | |
| AMX | 8/52 (15.38) | 9/108 (8.33) | 6/45 (13.33) | 0.366 |
| AMC | 6/52 (11.54) | 13/108 (12.04) | 6/45 (13.33) | 0.962 |
| CFM | 4/52 (7.69) | 8/108 (7.41) | 3/45 (6.67) | 0.980 |
| GEN | 5/52 (9.62) | 15/108 (13.89) | 5/45 (11.11) | 0.718 |
| TET | 5/52 (9.62) | 4/108 (3.70) | 1/45 (2.22) | 0.172 |
| DOX | 3/52 (5.77) | 3/108 (2.78) | 3/45 (6.67) | 0.482 |
| AZM | 19/52 (36.54) | 51/108 (47.22) | 21/45 (46.67) | 0.418 |
| CLR | 18/52 (34.62) | 50/108 (46.30) | 22/45 (48.89) | 0.283 |
| LVFX | 17/52 (32.69) | 32/108 (29.63) | 15/45 (33.33) | 0.872 |
| SPFX | 12/52 (23.08) | 21/108 (19.44) | 11/45 (24.44) | 0.749 |
| MTZ | 15/52 (28.85) | 52/108 (48.15) | 17/45 (37.78) | 0.059 |
| TID | 21/52 (40.38) | 56/108 (51.85) | 17/45 (37.78) | 0.185 |
| RFP | 2/52 (3.85) | 4/108 (3.70) | 5/45 (11.11) | 0.153 |
| FZD | 5/52 (9.62) | 13/108 (12.04) | 3/45 (6.67) | 0.599 |

Note:
CSG, chronic superficial gastritis; CAG, chronic astropic gastritis; PU, peptic ulcer; AMX, amoxicillin; AMC, amoxicillin and clavulanate potassium; CFM, cefixime; GEN, gentamicin; TET, tetracycline; DOX, doxycycline; AZM, azithromycin; CLR, clarithromycin; LVFX, levofloxacin; SPFX, sparfloxacin; MTZ, metronidazole; TID, tinidazole; RFP, rifampicin; FZD, furazolidone.

the serious challenge of high antimicrobial resistance of *H. pylori* decreases the efficiency of the first-line treatment. Understanding the prevalence of local antibiotic resistance can provide useful information for the selection of treatment strategies. To our knowledge, this is the first multicenter study in Fujian, China to evaluate antibiotic resistance profile of *H. pylori* to so many antibiotics. Recent guidelines recommend amoxicillin, clarithromycin and other five antibiotics. However it is important to seek novel and effective drugs for *H. pylori* eradication in an era of increasing antibiotic resistance.

In our study, we showed that the prevalence of *H. pylori* resistance to AZM, CLR, LVFX, SPFX, MTZ, and TID was high, whereas the prevalence of *H. pylori* resistance to AMX, AMC, CFM, TET, DOX, RFP, and FZD was relatively low in Fujian, China. Clarithromycin, metronidazole, and levofloxacin are the most commonly used antibiotics for *H. pylori* treatment; thus, their resistance has been widely studied. In our study, the resistance rates to the three antibiotics were all >30%, similar to other regions of China (*Bai et al., 2015*; *Su et al., 2013*), but higher than those in the Americas, Europe, and South-East Asia (*Savoldi et al., 2018*). CLR+MTZ, CLR+LVFX, and CLR+MTZ+LVFX were the main multiple resistance patterns; therefore, these three antibiotics should not be used simultaneously to treat *H. pylori*. Additionally, resistance rates to clarithromycin, metronidazole, and levofloxacin have increased over time in all WHO regions (*Savoldi et al., 2018*). This may be due to macrolide and quinolone consumption for other infections in the community. *Megraud et al. (2021)* found a significant association between *H. pylori* resistance to clarithromycin and community consumption of macrolides, and

between *H. pylori* resistance to levofloxacin and consumption of quinolones and second-generation quinolones.

According to the Maastricht VI/Florence Consensus Report (*Malfertheiner et al., 2022*), triple clarithromycin-containing therapy should only be used after determining the susceptibility of *H. pylori* to antimicrobials in regions with high rates of resistance to clarithromycin (>15%). Therefore, susceptibility testing for CLR before clarithromycin-containing therapy should be proposed in Fujian. Metronidazole can be used at increasing doses to overcome bacterial resistance, but its side effects also increase.

Both CLR and AZM are macrolide antibiotics, LVFX and SPFX are quinolone antibiotics, and MTZ and TID are nitroimidazole antibiotics. Our study demonstrated that the resistance profiles of AZM, SPFX, and TID were all as high as those of their peers. One reason for this may be the cross-resistance that exists in them. The other reason may be their broad use in treating other infections in Fujian. For example, AZM is a broad-spectrum therapeutic used to treat bronchitis, pneumonia, sexually transmitted diseases, and infections of the ears, lungs, sinuses, skin, throat, and reproductive organs (*Firth & Prathapan, 2020*).

Unlike CLR, LVFX, and MET, *H. pylori* resistance to TET, RFP, and FZD has remained low and stable in Fujian and worldwide in recent years (*Savoldi et al., 2018*). However, they are not readily available in local clinical practice.

Doxycycline (DOX) and minocycline are second-generation tetracycline antibiotics with similar chemical structures and antibacterial spectra; however, DOX is inexpensive. *Gu et al. (2019)* demonstrated that 14-day bismuth-containing quadruple therapy with amoxicillin and doxycycline achieved an eradication rate of 93.8% (105/112; 95% CI [89.2–98.3]) in the per-protocol analysis and 89.8% (106/118; 95% CI [84.3–95.4]) in the intention-to-treat analysis. A meta-analysis also reported that 7-day and 10-day quadruple therapy with doxycycline might be an optional and effective eradication regimen as first-line therapy against *H. pylori* infection in clinical practice (*Zhao et al., 2021*). Moreover, some studies have reported that DOX is more effective than antibiotic and has a number of non-antibiotic effects, such as anti-inflammatory, anti-oxidant, neuroprotective, immunomodulatory, and anti-apoptotic effect (*Gomes & Fernandes, 2007*; *O'Dell et al., 2006*; *Singh, Khanna & Kalra, 2021*). Therefore, DOX may be a new choice for *H. pylori* eradication in Fujian, and AMX and AMC are both members of penicillin. AMX is the core medicine for the eradication of *H.pylori* because *H. pylori* remains susceptible to AMX long-term, as demonstrated in our study and other studies (*Bai et al., 2015*; *Su et al., 2013*). In addition, AMX is widely available and has few adverse effects. However, a penicillin skin test should be conducted before the use of AMX, which is inconvenient for patients. Approximately 5–15% of people are allergic to penicillin or have a positive skin test (*Blumenthal et al., 2019*), therefore unsuitable for taking amoxicillin. Effective treatments for *H. pylori*-infected patients especially for those who are allergic to penicillin are still lacking.

Furazolidone and tetracycline were recommended by the Fifth Chinese National Consensus Report on the Management of *H. pylori* Infection for patients allergic to penicillin. Because they have low antibiotic resistance, but they are not easily available in

clinical practice, and the risk of side effects is also relatively high. Cefixime is a third generation cephalosporin. Cephalosporins and penicillin are β-lactam antibiotics that share a similar bactericidal mechanism (*Fagoonee et al., 2013*). *Fera, Carbone & Foca (1993)* evaluated the *in vitro* activities of 19 antimicrobial agents against 18 strains of *H. pylori*, and found that among cephalosporins, cefixime had the highest antibacterial activity. *In vitro* antimicrobial sensitivity testing demonstrated a low resistance rate for cephalosporins in *H. pylori* (similar to that of amoxicillin) (*Samra et al., 2002*). Our findings also showed that the resistance of *H. pylori* to cefixime was relatively low. A few clinical studies revealed that cefixime-containing regimens achieve relatively satisfactory eradication rates (*Fu et al., 2017*; *Tatsuta et al., 1990*).

In conclusion, the prevalence of *H. pylori* resistance to AZM, CLR, LVFX, SPFX, MTZ, and TID was high, whereas the prevalence of *H. pylori* resistance to AMX, AMC, GEN, CFM, TET, DOX, RFP, and FZD was relatively low in Fujian. CFM and DOX are promising new choices for *H. pylori* eradication.

### Funding
This research was jointly supported by the Natural Science Foundation of Fujian Province (Grant no. 2020J011087, 2022J01121025), the Medical Innovation Project of Fujian Provincial Health Commission (Grant no. 2020CXA006), the Zhuhai Science and Technology Project (Grant no. 20171009E030078), and the Startup Fund for Scientific Research of Fujian Medical University (Grant no. 2020QH1258). The funders had no role in study design, data collection and analysis, decision to publish, or preparation of the manuscript.

### Grant Disclosures
The following grant information was disclosed by the authors:
Natural Science Foundation of Fujian Province: 2020J011087 and 2022J01121025.
Medical Innovation Project of Fujian Provincial Health Commission: 2020CXA006.
Zhuhai Science and Technology Project: 20171009E030078.
Startup Fund for Scientific Research of Fujian Medical University: 2020QH1258.

### Competing Interests
The authors declare that they have no competing interests.

### Author Contributions
- Xueping Huang conceived and designed the experiments, performed the experiments, prepared figures and/or tables, authored or reviewed drafts of the article, and approved the final draft.
- Baihe Wu conceived and designed the experiments, analyzed the data, prepared figures and/or tables, authored or reviewed drafts of the article, and approved the final draft.
- Qiuzhao Chen performed the experiments, authored or reviewed drafts of the article, and approved the final draft.

- Yushan Chen performed the experiments, authored or reviewed drafts of the article, and approved the final draft.
- Xinhua Ji performed the experiments, authored or reviewed drafts of the article, and approved the final draft.
- Xiang Zhou performed the experiments, authored or reviewed drafts of the article, and approved the final draft.
- Biao Suo performed the experiments, authored or reviewed drafts of the article, and approved the final draft.
- Zhihui Lin conceived and designed the experiments, analyzed the data, authored or reviewed drafts of the article, and approved the final draft.
- Xiaoling Zheng conceived and designed the experiments, analyzed the data, authored or reviewed drafts of the article, and approved the final draft.

### Human Ethics

The following information was supplied relating to ethical approvals (*i.e.*, approving body and any reference numbers):

The Ethics Committee of the Fujian Provincial Hospital approved the study (Approval No. K2019-08-008).

### Data Availability

The raw measurements are available in the Supplemental File.

### Supplemental Information

Supplemental information for this article can be found online at http://dx.doi.org/10.7717/peerj.15611#supplemental-information.

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
