# Peer review of "Antibiotic resistance profile of Helicobacter pylori to 14 antibiotics: a multicenter study in Fujian, China"

_PeerJ, doi:10.7717/peerj.15611_

## Round 0.1 · original submission · Major Revisions

Please address the reviewers' concerns point by point.

·

Basic reporting

The language was clear and concise. Background could have been provided in a more detailed manner. Raw data was shared but the experimental data and the related figure legends could have been more clearly explained.

Experimental design

Antibiotics dosing and the related MIC's were not provided in the experimental designs. It is also very important to know what are the currently prescribed antibiotics that do not show resistance and how the study can improve the drug regimen for H.pylori infections need to be listed elaborately.

Validity of the findings

Data and statistical analysis are presented in the study and reasonably verified with replicates. But the sample size could have been bigger. Also, if the sample of patients had any underlying complications or are under any other antibiotics for any other infections has not been discussed.

Reviewer 2 ·

Basic reporting

I read with great interest the manuscript by Xueping Huang et al. This manuscript is timely as resistance in H. pylori infection has accumulated to become a major global public health problem. The WHO has rightly decided to rank H. pylori among the 20 infections that pose the greatest threat to human health due to their resistance to antimicrobials. At the origin of this problem, there is an intensive, extensive, and empirical use of the already very limited range of antibiotics acting on the species in addition to the extraordinary adaptive skills displayed by H. pylori. This situation currently imposes antimicrobial stewardship which increasingly implies not only an adjustment of the therapy to the local epidemiology of drug resistance but also a susceptibility antimicrobial test prior to any treatment. This article, therefore, touches on a topical issue.
Overall, the article is well-written and well-organized. The article is self-contained. There are a few typos in this manuscript and this is to the advantage of the authors. Nevertheless, I noted a striking resemblance between the text presented and another one published previously by the authors (https://doi.org/10.1371/journal.pone.0256225). Whole swathes of text (e.g., in the method section) seem to have simply been copied and pasted; I recommend that authors take care to keep each of their publications unique. The ethical clearances of that study and the current one are exactly the same. In the discussion, I note unreferenced assertions (e.g. lines 204 to 210). Sometimes “H. pylori” is not italicized (e.g. line 207). "Reproductive organ" (line 2019) seems to require an "s". On line 223, the authors should write “...in local clinical practice” rather than generalizing the situation by writing “...in clinical practice”. Likewise, the statement in line 242, on the absence of effective treatment must be relativized or contextualized. Abbreviations “PP analysis” and “ITT analysis” (line 229) should be explained at their first citation in the text.

Experimental design

I note a few problems in the experiment design that require clarification and corrections. Mainly, the authors used culture media which are unusual. The isolation of H. pylori was thus performed from the biopsies using brain-heart broth with 10% calf serum. This medium being non-selective, I wonder how the authors could handle other micro-organisms which may grow together with H. pylori on such non-selective medium, were managed. Then, the antimicrobial susceptibility test (AST) was carried out by the broth microdilution method. Although significant agreement of results is reported with agar dilution ,
The EUCAST criteria applied by the authors are not intended for the interpretation of the results obtained with this method but with the agar dilution method (i.e. the standard method for AST of H. pylori, generally advocated by organizations such as the CLSI). This should be acknowledged as a weakness of this study even if a discussion can be made on a possible (but not perfect) agreement of result obtained with the two methods (https://doi.org/10.1093/infdis/jiac389). More importantly, the authors claim to refer to EUCAST for the interpretation of their results; yet the resistance breakpoints used (lines 134 to 139) differ from those recommended. Indeed, the EUCAST recommends 0.125 mg/L as a breakpoint for AMX (instead of 2 mg/L used here), 0.25 mg/L for CLA (instead of 0.5 mg/L used), and 1mg/L of TET (instead of 4 mg/L used). In addition, the authors attribute to EUCAST the resistance breakpoints used for AZM, TID, SPFX, GEN, CFM, and AMC. however, these drugs are not included in the EUCAST's recommendations for H. pylori. The real source of the criteria applied should be provided. For reference on the EUCAST, visit the page 83 of the EUCAST Clinical Breakpoint Tables v. 13.0 (published on 2023 -01-01; https://www.eucast.org/fileadmin/src/media/PDFs/EUCAST_files/Breakpoint_tables/v_13.0_Breakpoint_Tables.pdf).

Validity of the findings

It therefore clearly appears that the rates of resistance reported in the study have probably been underestimated in the current study which relied on higher breakpoints than those recommended by the EUCAST. This could have potentially undermined the validity of the conclusions drawn by the authors.

Additional comments

No additional comments

---

## Round 0.2 · accepted · Accept

Both reviewers and I think your manuscript is ready for publication.

·

Basic reporting

The language is clear and concise. References were properly cited throughout the article. Hypotheses were addressed and the results were correlated well.

Experimental design

All the criteria were met and there are no major flaws.

Validity of the findings

Conclusions were presented well and the article has a significant impact in the field of H.pylori drug resistance research.

Reviewer 2 ·

Basic reporting

I thank Huang X et al. for clarification of issues raised in my last reviewing work. This manuscript has kept its relevance and topicality with regard to the current global situation of the H. pylori infection treatment that I had noted previously. Other regions of the world should be inspired to document the local susceptibility of strains of H. pylori to best guide clinical practice. I am convinced that many researchers and clinical practitioners would be interested in reading this study. While reworking the text, the authors kept their nice writing and the ideas organized throughout the manuscript. The few typos initially noted have been tracked through the text and corrected. The text now has an original character with regard to the authors' previous publications, including the one under the same project. The misquoting of ethical clearances has been corrected. Additional references have been added to support the discussion within the manuscript. The issues relating to the experimental design and the validity of the results are, in my opinion, now clarified. I find that the manuscript is satisfactory at the present stage.

Experimental design

No comment

Validity of the findings

No comment

Additional comments

No comment